# Cost-Effectiveness Analysis of HPV Extended versus Partial Genotyping for Cervical Cancer Screening in Singapore

**DOI:** 10.3390/cancers15061812

**Published:** 2023-03-16

**Authors:** Brandon Chua, Li Min Lim, Joseph Soon Yau Ng, Yan Ma, Hwee Lin Wee, J. Jaime Caro

**Affiliations:** 1Saw Swee Hock School of Public Health, National University of Singapore, 12 Science Drive 2, Singapore 117549, Singapore; e0639551@u.nus.edu or brandon.chua@bd.com (B.C.); jaime.caro@mcgill.ca (J.J.C.); 2Health Economics and Outcomes Research, Becton Dickinson Holdings Pte. Ltd., 2 International Business Park Road, The Strategy #08-08, Singapore 609930, Singapore; viva.ma@bd.com; 3Division of Gynaecologic Oncology, Department of Obstetrics and Gynecology, National University Hospital, 5 Lower Kent Ridge Rd., Singapore 119074, Singapore; li_min_lim@nuhs.edu.sg (L.M.L.); obgnsyj@nus.edu.sg (J.S.Y.N.); 4Department of Pharmacy, National University of Singapore, 18 Science Drive 4, Singapore 117543, Singapore; 5School of Global and Population Health, McGill University, Suite 1200, 2001 McGill College Avenue, Montréal, QC H3A 1G1, Canada; 6Department of Health Policy, London School of Economics and Political Science, Houghton Street, London WC2A 2AE, UK; 7Evidera, 500 Totten Pond Rd., Waltham, MA 02451, USA

**Keywords:** HPV extended genotyping, cost-effectiveness, economic evaluation, cervical cancer screening, Singapore, DICE simulation

## Abstract

**Simple Summary:**

Identification of human papillomavirus (HPV) genotypes beyond HPV16 and HPV18 by extended genotyping (XGT) for cervical cancer screening, allows for risk stratification for clinical management. We estimated the resource use, cost, and quality-adjusted life years (QALY) comparing XGT to partial genotyping (PGT) in Singapore. The analysis considers a five-year screening cycle and lifetime outcomes in women diagnosed with CIN2+. Compared to PGT, XGT cost an additional SGD 16,370 per QALY gained (USD 19,465 per QALY gained), with 7130 (19.4%) fewer colposcopies, 6027 (7.0%) fewer cytology tests, 9787 (1.6%) fewer clinical consultations, yet 2446 (0.5%) more HPV tests. XGT can be a cost-effective, risk-based approach to primary cervical cancer screening as it utilizes fewer resources compared to PGT. This analysis, not previously conducted in an Asian context, could guide the use of XGT in Asia.

**Abstract:**

Human papillomavirus (HPV) partial genotyping (PGT) identifies HPV16 and HPV18 individually, alongside 12 other high-risk HPV genotypes (hrHPV) collectively. HPV extended genotyping (XGT) identifies four additional hrHPV individually (HPV31, 45, 51, and 52), and reports the remaining eight in three groups (HPV33|58; 56|59|66; 35|39|68). Quality-adjusted life years (QALY), health care resource use, and costs of XGT were compared to PGT for cervical cancer screening in Singapore using DICE simulation. Women with one of the three hrHPV identified by XGT (HPV35|39|68; 56|59|66; 51), and atypical squamous cells of undetermined significance (ASCUS) on cytology, are recalled for a repeat screening in one year, instead of undergoing an immediate colposcopy with PGT. At the repeat screening, the colposcopy is performed only for persistent same-genotype infections in XGT, while with PGT, all the women with persistent HPV have a colposcopy. Screening 500,122 women, aged 30–69, with XGT, provided an incremental cost-effectiveness ratio (ICER) versus PGT of SGD 16,370/QALY, with 7130 (19.4%) fewer colposcopies, 6027 (7.0%) fewer cytology tests, 9787 (1.6%) fewer clinic consultations, yet 2446 (0.5%) more HPV tests. The XGT ICER remains well below SGD 100,000 in sensitivity analyses, (-SGD 17,736/QALY to SGD 50,474/QALY). XGT is cost-effective compared to PGT, utilizes fewer resources, and provides a risk-based approach as the primary cervical cancer screening method.

## 1. Introduction

In 2020, over 600,000 women were diagnosed with cervical cancer globally, while 300,000 died from it [1]. In a call to eliminate cervical cancer, the World Health Organization (WHO) has launched the 90-70-90 target for countries to achieve by 2030: (1) 90% of girls fully vaccinated against human papillomavirus (HPV) by the age of 15; (2) 70% of women screened with a high-performance test by the age of 35 and 45; (3) 90% of women identified with cervical precancer and cancer received treatment [2].

HPV is responsible for over 90% of cervical cancers and 14 high-risk HPV genotypes (hrHPV) have been identified [2]. Although HPV66 has been reclassified as “possibly oncogenic” by the International Agency for Research on Cancer [3], many validated HPV genotyping assays continue to identify it [4]. The WHO recommends HPV DNA testing as the first-choice screening modality for cervical cancer prevention [2], and a trend toward primary HPV screening has been observed globally [5]. HPV genotyping technologies are classified based on the individual identification of genotypes [6]. Partial genotyping (PGT) individually identifies HPV16 and HPV18, while grouping the remaining 12 hrHPV. Extended genotyping (XGT) individually identifies four further hrHPV (e.g., HPV31, 45, 51, and 52), with the remaining eight reported in various combinations (e.g., HPV33|58; 56|59|66; 35|39|68) [7]. 

The elevated risk of a diagnosis of cervical intraepithelial neoplasia grade 3 or worse (CIN3+) with HPV16/18 is well established [2,8]. Recent evidence suggests that the risk with the 12 other hrHPV varies across a wide range: with HPV31 (7.9−8.9%) and HPV33 (5.4%−15%) having higher risks than HPV18 (2.7−9%) [8], thus, warranting a colposcopy. HPV35/39/68/51/56/59/66 with a low-grade squamous intraepithelial lesion (LSIL) or atypical squamous cells of undetermined significance (ASCUS), possess a low risk (2.0%) [8]. For patients with these genotypes, unnecessary coloscopies can be avoided. Repeat screening a year later would reserve colposcopy for women with a persistent same-genotype infection (PSGI), which accounts for half of all persistent infections, and carries a higher risk of CIN3+, when compared to those with a change in HPV genotype [9]. 

Identifying hrHPV beyond HPV16/18 is a critical strategy to eliminate cervical cancer [10,11,12], because increasing HPV vaccine coverage has shifted the HPV epidemiology [13,14]. In Australia, school-based HPV vaccinations reduced HPV16/18 prevalence to 2.3% in 2021, while the prevalence of the other hrHPV remains high (8.8%) [15]. High-grade cervical lesions attributable to non-vaccine preventable genotypes may increase among vaccinated people. In the Costa Rica vaccine trial with seven years of follow-up, women vaccinated against HPV16/18 had 9.2 per 1000 additional cervical intraepithelial neoplasia grade 2 or worse diagnoses (CIN2+) attributable to non-vaccine preventable genotypes (HPV35/39/51/52/56/58/59), compared to unvaccinated women [16]. Hence, the value of screening hrHPV beyond HPV16/18 is expected to increase as countries progress with their cervical cancer elimination targets [12]. 

There are geographical differences in specific hrHPV that contribute to invasive cancers. In Asia, HPV52/58 is more prevalent compared to other regions of the world [17]. In Singapore, HPV52/33/58 are among the five most prevalent hrHPV associated with cervical cancer [18]. Yet, the national screening program uses PGT, which provides no information on these HPV genotypes. In an interview with Singapore healthcare providers, the value of specific HPV genotyping beyond HPV16/18 was highlighted [19]. For example, with XGT, the prevalence of specific non-HPV16/18 genotypes can be established, and those that pose a lower risk can be identified for less intensive management [19]. Based on the above, we undertook a systematic evaluation of the benefits and consequences of using XGT in the national screening program. While the specific inputs pertain to Singapore, the methods and model should be of general interest.

## 2. Materials and Methods

A discretely integrated condition event (DICE) simulation was constructed in Microsoft Excel 365 (Redmond, WA, USA) to evaluate the cost and outcomes of XGT compared to PGT, from a healthcare payer’s perspective. In DICE, information is stored as “conditions” and changes in information are triggered by “events” at specific time points during the simulation. These changes are processed using a general Visual Basic for Applications macro [20,21]. With DICE, the design and implementation of models can be simplified and kept very transparent [20]. It has been used in many economic evaluations, including breast cancer chemoprevention [20], gene therapy in thalassemia major [22], rotavirus vaccinations [23], tardive dyskinesia treatment [24], rheumatoid arthritis treatment [25], and management of COVID-19 [26]. The engine, templates, and manuals are available from https://dice.impact-hta.eu/.

A cohort of 500,122 screening candidates was modeled. The model considers unvaccinated screening candidates aged 30 to 69 years old [27,28]. No vaccination was assumed because the school-based HPV vaccination program only began in Singapore in 2019 [29]. HPV screening was part of the initial screening test. Screening coverage was estimated at 48.2% [30]. The model considers one five-year screening cycle and lifetime outcomes if diagnosed with CIN2+. All assumptions are detailed in Appendix A. The model was designed, populated, and analyzed in line with the ISPOR-SMDM Good Research Practice Guidelines [31].

### 2.1. Model Schematics and Screening Algorithm

Screening algorithms with PGT and XGT are shown in Figure 1. With PGT, management differed if it was HPV16/18 or not, according to the Society for Colposcopy and Cervical Pathology of Singapore (SCCPS) guidelines [32]. Colposcopy was recommended for HPV16/18. In all other hrHPV, reflex cytology was performed, and those with ASCUS or worse (ASCUS+) were scheduled for a colposcopy. If negative for intraepithelial lesion or malignancy (NILM), repeat screening in one year was recommended. At the repeat screening, all women still positive for HPV, regardless of genotype, were scheduled for a colposcopy. 

With XGT, management was the same as with PGT for HPV16/18, and the 12 other hrHPV with LSIL+ or NILM. For ASCUS, management depended on additional risk grouping: (1) Group A (HPV31/33/45/52/58) went for a colposcopy; (2) Group B (HPV35/39/51/56/59/66/68) was scheduled for repeat screening owing to a lower risk of CIN2+ [8]. Hence, group B genotypes with ASCUS were assumed to have a similar risk as non-HPV16/18 with NILM. At the repeat screening, one year after the initial screening, PSGI led to a colposcopy, while a change in the HPV genotype was managed as a new infection. A colposcopy was also scheduled for those who remained HPV positive after two repeat screenings (i.e., a persistent HPV infection for three years), regardless of the genotype. 

The post-colposcopy follow-up was modeled (Figure 2) by SCCPS guidelines [32], together with expert clinical opinions, and assumed: (1) CIN negative or CIN1 with ASCUS or LSIL at the one-year follow-up was managed similarly to NILM; (2) all CIN2/3 diagnoses were treated, and no recurrence occurred, as it is expected to be low [33].

### 2.2. Model Inputs

Singapore-specific data were prioritized for model inputs (Table 1), supplemented with published data from elsewhere or expert opinions from Singapore clinicians. Cost inputs were obtained from a recent study evaluating the cost-effectiveness of the HPV vaccine, conducted by the Agency of Care Effectiveness (ACE), the national health technology assessment agency in Singapore [34]. The XGT test fee charged by providers was assumed to be 15% above that for the PGT. All other cost inputs did not differ between PGT and XGT. Cost inputs for cancer treatment were weighted by cancer stage. All costs were reported in 2020 Singapore dollars (SGD) and converted to US dollars (USD) using the 2020 purchasing power parity (SGD0.841 = USD1) [35].

Utility values were based on US estimates for cervical cancer, precancer, and screening [36,37], which was the basis for the economic evaluation of HPV vaccines in Singapore [34]. Disutility values were derived by subtracting utility values from one. Disutility values for cancer treatment and survivors were weighted by cancer stage. Quality-adjust life year (QALY) loss due to treatment or screening was calculated by multiplying disutility values by the duration of the procedure (available in Appendix A). For example, assuming the disutility of screening lasted two weeks, the QALY loss from screening was 
$0.02\times \frac{2}{52}=0.0007$
. For deaths, QALY loss was equivalent to life expectancy, according to the Singapore life table [38]. QALY loss among cancer survivors was assumed to be similar to that without cancer after 10 years of treatment [39]. Hence, the average lifetime QALY loss with cancer was derived using 10-year survivals for cervical cancer [40], life expectancy [38], and cancer survivor disutility values [34].
cancers-15-01812-t001_Table 1Table 1Inputs for the model.InputBase CaseLower LimitUpper LimitDistributionReferenceNumber eligible1,037,598---[27,28]Screening coverage48.2%45.8%50.7%Beta[30]Follow-up non-adherence * 25.0%0%40%-[28] ^†^
**Clinical inputs**
hrHPV




 Prevalence9.2%7.9%10.5%Beta[28] % non-HPV16/1880.8%70.3%83.0%Beta[28]  % Group B 56.6%51.0%61.0%Beta[41]  % NILM56.1%---[42]ASCUS among:




 Group B 31.8%---[42] Group A 40.6%---[42]CIN1 regressing in 1 year60.0%45.0%73.0%Beta[43]Cancers among:




 CIN2+ diagnosis2.6%2.3%2.9%Beta[44] CIN2+ of Group B with ASCUS2.6%0.0%10.0%-[44,45,46,47]CIN2+ risk with:




 Group B with ASCUS6.1%2.6%9.5%Beta[45,46,47,48] Group A with ASCUS14.2%---[48] HPV16/1821.9%---[48] Non-HPV16/18 with LSIL+16.4%---[48]PSGI at repeat screening57.1%54.2%60.1%Beta[49,50,51]hrHPV 1 yr persistence43.3%41.8%44.8%Beta[52]HSIL/ASC-H 1 year post CIN1/negative for CIN6.7%5.7%7.7%Beta[53]ASCUS+/HPV+ 2 years post CIN1/negative for CIN15.4%13.8%17.1%Beta[28]Proportion stage I cancer40.8%---[54]Proportion stage II cancer24.4%---[54]Proportion stage III cancer18.1%---[54]Proportion stage IV cancer16.7%---[54]10-year cancer survival45.4%---[40]XGT repeat screenings215-
^†^
Annualized CIN2+ risk for 




HPV genotype persistence




 Same5.7%---[55] Change1.9%---[55] Regardless of genotype3.3%---[55] Multiplier for CIN2+ risk10.71.38Normal
^†^
Annualized CIN2+ risk for CIN1/negative for CIN




 1 negative pap smear1.1%---[56] ASCUS/LSIL upon follow-up2.1% ---[56] ASC-H upon follow-up5.3%---[56] HSIL+ upon follow-up3.4%---[56]
**Cost inputs SGD (USD)**
Clinic visit75 (89)37 (44)113 (134)Normal[34]Cytology79 (94)39 (46)119 (141)Normal[34]HPV DNA (PGT)115 (137)57 (68)173 (206)Normal[34]CIN2/3 treatment3662 (4354)1832 (2178)5492 (6530)Normal[34]Colposcopy350 (416)174 (207)526 (625)Normal[34]Biopsy500 (595)250 (297)750 (892)Normal[34]Colposcopies with biopsies8%---
^†^
Stage I cancer treatment28,350 (33,710)14,176 (16,856)42,524 (50,564)-[34]Stage II cancer treatment34,568 (41,103)17,284 (20,552)51,852 (61,655)-[34]Stage III cancer treatment34,568 (41,103)17,284 (20,552)51,852 (61,655)-[34]Stage IV cancer line 1 treatment43,016 (51,149)21,508 (25,574)64,524 (76,723)-[34]Stage IV cancer line 2 treatment 75,552 (89,836)37,776 (44,918)113,328 (134,754)-[34]Cancer treatment ^‡^37,227 (44,265)29,781 (35,412)44,672 (53,118)NormalCalculatedXGT cost factor1.151.001.30-
^†^

**Utility**
Screening0.9800.9700.990-[34]Colposcopy normal results0.9500.9240.976-[34]CIN1 0.9100.8880.954-[34]CIN2/3 0.8700.8040.936-[34]Cancer Stage I0.6500.4900.810-[34]Cancer Stage II/III0.5600.4200.700-[34]Cancer Stage IV0.4800.3600.600-[34]Cancer stage I survivor0.9700.7300.990-[34]Cancer stage II/III survivor0.9000.6800.990-[34]Cancer stage IV survivor0.6200.4700.780-[34]
**QALY loss**
Screening0.0007690.0003840.00115NormalCalculatedCIN1 or negative for CIN10.005380.002690.00723NormalCalculatedCIN2/30.02000.009850.0302NormalCalculatedCancer treatment ^‡^0.09300.06400.121NormalCalculatedAverage lifetime QALY loss for cancer ^‡^18.714.922.4NormalCalculated* repeat screening, post-colposcopy for CIN1/negative for CIN; ^†^ assumption; ^‡^ weighted by stage Abbreviations: ASCUS: atypical squamous cells of undetermined significance; ASC-H: atypical squamous cells cannot exclude high-grade squamous intraepithelial lesion; CIN: cervical intraepithelial neoplasia; hrHPV: human papillomavirus genotypes; HSIL: high-grade squamous intraepithelial lesion; PGT: HPV partial genotyping; NILM: negative for intraepithelial lesion or malignancy; XGT: HPV extended genotyping.

### 2.3. Cost-Effectiveness Analysis

The total cost and QALY loss with XGT were compared to PGT. The difference in costs divided by the difference in QALY loss provided the incremental cost-effective ratio (ICER). All costs and QALY losses were discounted at 3.0% annually, in line with ACE recommendations [57]. The cost-effectiveness threshold was taken as SGD 100,000 (USD 118,906), comparable to the gross domestic product per capita in Singapore in 2021 (SGD 97,798) [58].

In deterministic one-way sensitivity analysis inputs were varied across their corresponding 95% confidence intervals (CI), or by ±20% when CI was unavailable. Probabilistic sensitivity analysis (PSA) was conducted with 1000 simulations. The range of inputs and distributions used for the uncertainty analysis are detailed in Table 1. 

Three scenario analyses were conducted. The effect of detecting cancers for group B genotypes with ASCUS in the initial XGT screening was simulated by assuming a 20% higher cost and QALY loss, estimated from the treatment of later-stage cancers [34]. Second, alternative inputs from China were used for the CIN2+ risk and proportion of cancers diagnosed among those in group B with ASCUS [45,46,47]. Third, the effect of different HPV burdens and distribution of HPV genotypes due to HPV vaccination was examined by varying HPV prevalence, the proportion of non-HPV16/18 infections, and the proportion of group B genotypes among non-HPV16/18 infections. Data from Portugal was used, which represented a setting with a high population coverage (>80%) of HPV vaccinations since 2008, with individual reporting of all 14 hrHPV for screening since 2016 [59]. 

### 2.4. Validation and Verification

Face validity was ascertained by experts on the team with expertise in gynecologic oncology. External validation was not conducted given the lack of national data on cervical cancer incidence and mortality by screening status in Singapore. The implementation of the DICE model was verified by an independent modeler. Further, the model execution was verified by inspecting the DICE model text log, which specified every instruction executed together with its result. 

## 3. Results

In the base case analysis (Table 2), the use of XGT resulted in 7130 (19.4%) fewer colposcopies, 9787 (1.6%) fewer clinic consultations, and 6027 (7.0%) fewer cytology tests over a 5-year period; yet an additional 2446 (0.5%) HPV tests, giving 274.42 more QALYs, at an additional cost of SGD 4,492,120 (USD 5,341,403) over a 5-year period, compared to PGT. This yields an ICER of SGD 16,370/QALY (USD 19,465/QALY). Hence, XGT was cost-effective compared to PGT, at a willingness-to-pay threshold of SGD 100,000/QALY (USD 118,906/QALY). 

XGT was cost-effective across input ranges in one-way uncertainty analysis, with ICERs ranging between -SGD 17,736/QALY and SGD 50,474/QALY (-USD 21,089/QALY and USD 60,017/QALY) (Figure 3). The most important factor was the cost of XGT. Results were not particularly affected by QALY loss inputs, follow-up adherence, or the number of repeat screenings modeled for XGT. XGT was also cost-effective in all 1000 simulations in PSA and in all scenario analyses (SGD 3822/QALY to SGD 60,209/QALY, equivalent to USD 4545/QALY to USD 71,592/QALY). Full details of the scenario analyses can be found in Appendix A

## 4. Discussion

To our knowledge, this is the first economic evaluation comparing XGT to PGT in the Asian context, where data inputs from Singapore and Asia were prioritized. We simulated the impact of persistent HPV infections, where PSGI and a change in HPV infections were managed differently. We also modeled post-colposcopy processes in detail and did extensive scenario analyses. Findings from this study suggest that using XGT is cost-effective for the national cervical cancer screening program in Singapore. Our findings are consistent with previous research in the US [60], where significant resource savings were also observed for colposcopy, cytology, and treatment of CIN2/3. We found a greater reduction in colposcopy (19.4% vs 9.5%) because only PSGI required colposcopy, while those with a change in genotype were managed as a new HPV infection.

We found the use of XGT to be cost-effective compared to PGT. This remains, even considering epidemiological shifts in HPV due to vaccination. With HPV vaccination, the proportions of high-grade lesions and abnormal cytology has decreased [61]. The positive predictive value of high-grade cytology for CIN2+ was also reduced among vaccinated women [62]. Although we did not vary cytology distribution by histology in this study, a lower proportion of cytological abnormalities associated with HPV vaccination is expected to favor XGT, as patients can be better managed based on genotype-specific risks. Furthermore, CIN2+ attributable to non-vaccine genotypes may increase with vaccinations, as previously observed in the Costa Rica vaccine trial [16]. Hence, with the growing public health efforts on HPV vaccination, risk stratification for immediate or future risk of the disease using XGT should continue to guide decisions on colposcopies [63,64].

Healthcare worker shortages and burnout are significant issues for healthcare systems globally [65]. With XGT, the utilization of healthcare resources can be substantially reduced, conferring benefits to both patients and healthcare systems. For nearly 15% of all women attending screenings (group B genotypes with ASCUS) and 50% with persistent HPV infections, an immediate colposcopy is not required. These will improve the efficiency and decision-making in healthcare systems, free up capacities, and help alleviate burnout in healthcare workers. For instance, in facilities having limited capacities for colposcopies, patients with the highest risk of CIN2+ can be prioritized based on results provided by XGT, but not PGT. 

Importantly, women under work and family caregiving pressures have left or considered leaving the workforce [66]. Many could be spared the anxiety, sexual dysfunction, and impaired quality of life associated with colposcopy [67,68]. Although fewer colposcopies with XGT may result in less CIN2+ treatment, the consequences are not likely to be significant due to the slow progression of CIN2/3 over a year [43]. In scenario analyses, XGT continued to be cost-effective after accounting for the impact of potentially missed cancers among group B genotypes with ASCUS. However, the potential impact of delaying colposcopies for group B genotypes with ASCUS should be further evaluated with real-world evidence from registry-based studies.

XGT is a promising advancement for HPV screening [69,70,71]. Based on current evidence on the risk of PSGI and non-HPV16/18 infections [8,9,72], XGT can better identify patients not needing a colposcopy or more intensive follow-ups compared to PGT. Additionally, XGT offers greater flexibility in monitoring the prevalent hrHPV of interest, which can evolve as evidence of hrHPV emerges. Of note, HPV52 and HPV58 are more commonly associated with precancers and cancers in Asia [17], while HPV35-associated cervical disease is more common among women of African descent [73]. Additionally, hrHPV may be reclassified to a lower-risk category along with HPV66 [3]. To date, however, XGT is not widely implemented in Asia, and experience with the technology is limited to western countries such as the U.S. [74], Canada [75], Denmark [76], and Italy [77]. For the wider adoption of XGT, it is critical to drive awareness of the higher risk of non-HPV16/18 infections and PSGI, which has been well documented [8,9,72]. This evidence can be incorporated into risk-based guidance for patient management. Risk-based guidance has been developed by the American Society for Colposcopy and Cervical Pathology [78], and guidance on genotype-specific management can be expected. Given differences in genotype-specific cervical cancers and precancers across regions [17], HPV screening algorithms may require local adaptation in national guidelines based on real-world data. 

This study has several limitations. First, the effect of multiple HPV coinfections, which may confer a higher risk of disease [79,80], was not modeled. Second, PSGI was not stratified further into risk groups, as with the initial HPV infection, because data on HPV distribution of PSGI and genotype-specific CIN2+ risk were not available. For example, persistent HPV52 (group A) and persistent HPV51 (group B) were assumed to have the same CIN2+ risk. Hence, HPV genotypes with a higher risk of CIN2+, such as HPV31, which should have an immediate colposcopy, were also not evaluated. Future studies can explore the impacts of different screening algorithms, when evidence on the genotype-specific risk for CIN2+ or CIN3+ stratified by cytology and histology is available. Third, utility inputs utilized for this study were from the U.S., which may not reflect how health statuses are valued in Singapore. However, our uncertainty analysis showed results that were not very sensitive to QALY loss inputs. Lastly, we did not model cancer stages in detail since PGT and XGT would not influence cancer progression differently. While this may overestimate the cost and QALY loss of cancer diagnosis and treatment, it does not significantly influence the study results based on uncertainty and scenario analyses.

## 5. Conclusions

In Singapore, the use of XGT can become the next primary cervical cancer screening method, providing a cost-effective and risk-based approach. Even with improved HPV vaccination coverage over time, XGT should remain valuable in stratifying patients for management based on risk profiles. Analysis of real-world data to support locally appropriate HPV testing algorithms, including XGT, will further improve the robustness of the national screening guidelines and the outcomes for women.

## Figures and Tables

**Figure 1 cancers-15-01812-f001:**
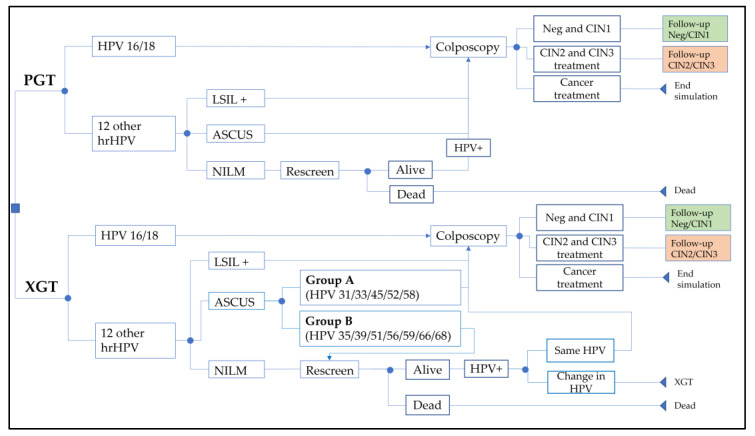
Screening algorithms modeled for PGT and XGT. Note, those HPV-negative on their rescreen resumed regular screening (5 years later). At ‘end simulation’, expected lifetime costs and QALYs are applied. Abbreviations: ASCUS: atypical squamous cells of undetermined significance; CIN: cervical intraepithelial neoplasia; hrHPV: high-risk human papillomavirus genotypes; LSIL: low-grade squamous intraepithelial lesion; PGT: HPV partial genotyping; NILM: negative for intraepithelial lesion or malignancy; Neg: negative for CIN; XGT: HPV extended genotyping.

**Figure 2 cancers-15-01812-f002:**
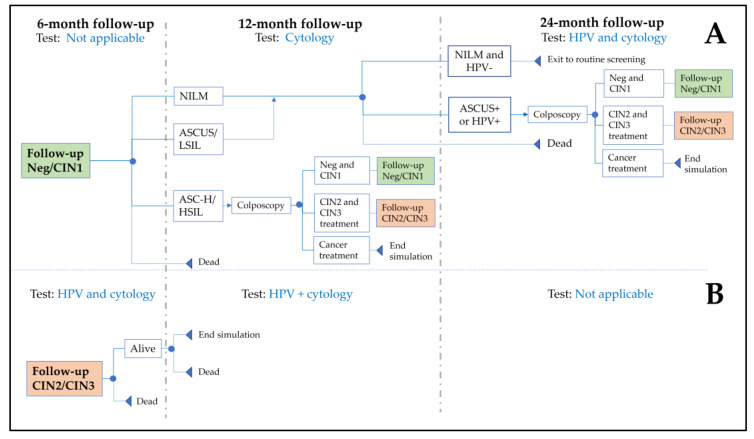
Model schematics of follow-up after colposcopy for (**A**) CIN1 or negative for CIN; (**B**) CIN2 or CIN3. At ‘end simulation’, expected lifetime costs and QALYs were applied. Abbreviations: ASCUS: atypical squamous cells of undetermined significance; ASC-H: atypical squamous cells cannot exclude high-grade squamous intraepithelial lesions; CIN: cervical intraepithelial neoplasia; HPV: human papillomavirus; HSIL: high-grade squamous intraepithelial lesion; LSIL: low-grade squamous intraepithelial lesion; PGT: HPV partial genotyping; NILM: negative for intraepithelial lesion or malignancy; Neg: negative for CIN; XGT: HPV extended genotyping.

**Figure 3 cancers-15-01812-f003:**
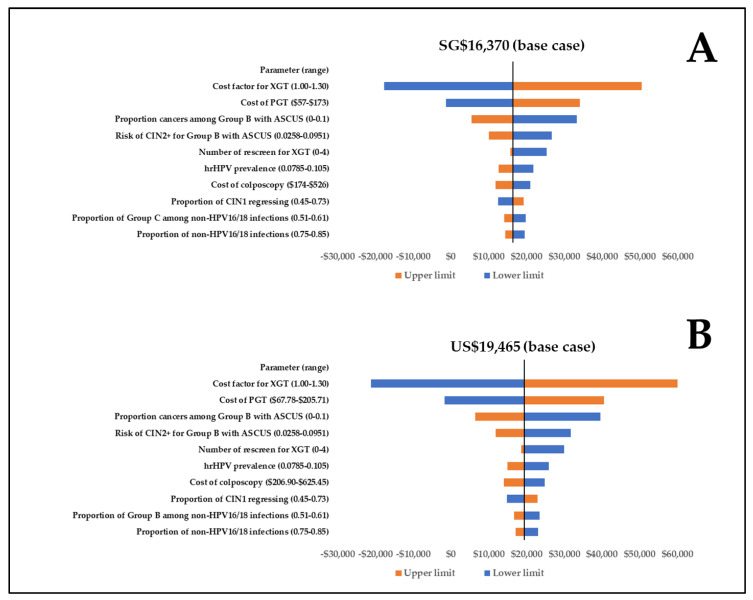
One-way uncertainty analysis for the ten most sensitive data inputs in (**A**) Singapore dollars; (**B**) US dollars. Abbreviations: ASCUS: atypical squamous cells of undetermined significance; CIN: cervical intraepithelial neoplasia; hrHPV: high-risk human papillomavirus genotypes; PGT: HPV partial genotyping; NILM: negative for intraepithelial lesion or malignancy; Neg: negative for CIN; XGT: HPV extended genotyping.

**Table 2 cancers-15-01812-t002:** Resource use, costs, and QALY for screening 500,122 candidates over five years.

Outcomes	PGT	XGT	Incremental
Cancers treated	106	98	−8
CIN2/3 treated	3993	3701	−292
CIN2+ treated	4099	3799	−300
Colposcopies conducted	36,809	29,679	−7130
HPV tests conducted	540,799	543,245	2446
Cytology conducted	86,419	80,392	−6027
Clinic consultations	601,796	592,009	−9787
Cost of HPV tests	SGD 62,106,581 (USD 73,848,491)	SGD 71,753,834 (USD 85,319,660)	SGD 9,647,253 (USD 11,471,169)
Cost of cytology	SGD 5,953,061 (USD 7,078,551)	SGD 5,658,456 (USD 6,728,247)	−SGD 294,605 (−USD 350,303)
Cost of colposcopy	SGD 14,259,917 (USD 16,955,906)	SGD 11,486,606 (USD 13,658,271)	−SGD 2,773,311 (−USD 3,297,635)
Cost of clinic consultation	SGD 45,038,254 (USD 53,553,215)	SGD 44,310,726 (USD 52,688,140)	−SGD 727,528 (−USD 865,075)
Total cost	SGD 145,904,751 (USD 173,489,597)	SGD 150,396,871 (USD 178,831,000)	SGD 4,492,120 (USD 5,341,403)
Total QALY loss	−6528.3	−6253.88	274.42
**ICER**	-	-	SGD 16,370/QALY (USD 19,465/QALY)

Abbreviations: CIN: cervical intraepithelial neoplasia; HPV: human papillomavirus; ICER: incremental cost-effectiveness ratio; PGT: HPV partial genotyping; QALY: quality-adjusted life years; XGT: HPV extended genotyping.

## Data Availability

No new data were created or analyzed in this study. Data sharing is not applicable to this article.

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
