# Peer review of "Cost-Effectiveness Analysis of HPV Extended versus Partial Genotyping for Cervical Cancer Screening in Singapore"

_cancers, 2023, doi:10.3390/cancers15061812_

Round 1

Reviewer 1 Report

The article addresses an important issue for countries with a high incidence of cervical cancer and low HPV vaccination rate. Cost-effectiveness of screening methods is also extremely important for countries with limited resources. I consider this an valuable, interesting and well-written article, of high scientific quality, suitable for publication. I have no other comments for the authors.

Reviewer 2 Report

The authors present a well-designed cost-effectiveness model on the impact of moving from partial HPV genotyping to extended genotyping within a cervical cancer screening activity in Singapore. The conclusions of this analysis are well based on the data analyzed and they provide a strong message on the potential benefits of the extended genotype. I have only minor comments for the authors to consider:

1.- In the introduction, it is described that ´ Extended genotyping (XGT) identifies four further hrHPV individually (HPV31, 45, 51, 52), with the remaining eight reported in three groups (e.g. HPV33|58; 56|59|66; 35|39|68) ´. It is important to state that this grouping could be refined.  For example, the inclusion of HPV66 as an oncogenic type has been questioned and the IARC classification did, remove HPV 66 as a high-risk oncogenic type. The impact is to screen as positive women that are unlikely to have the disease. Also, HPV35 accounts for 10% of cervical cancer cases in women with African ancestors. Although this may not have much relevance for the Asian population if an assay is produced for global use, HPV 35 should be considered as a higher oncogenic risk.

2.- In the costing parameters XGT is estimated to be in the base case scenario 1.15 compared to the PGT. Today extended genotype assays are available at no extra cost when compared to those only providing PGT.

3.- In the discussion, the statement ´CIN2+ attributable to non-vaccine genotypes is expected to increase with vaccination´ should be taken carefully as it has not been yet demonstrated that there will be an increase in the incidence of cases attributable to non-vaccine types  
